

# A pressure-driven atmospheric boundary layer model satisfying Rossby- and Reynolds number similarity

Maarten Paul van der Laan[1], Mark Kelly[1], and Mads Baungaard[1]

[1]DTU Wind Energy, Technical University of Denmark, Risø Campus, Frederiksborgvej 399, 4000 Roskilde, Denmark

**Correspondence:** Maarten Paul van der Laan (plaa@dtu.dk)

**Abstract.** Idealized models of the atmospheric boundary layer (ABL) can be used to leverage understanding of the interaction between the ABL and wind farms, towards improvement of wind farm flow modelling. We propose a pressure-driven one-dimensional ABL model without wind veer, which can be used as an inflow model for three-dimensional wind farm simulations for isolating the effects of wind veer and ABL depth. The model is derived from the horizontal momentum equations, and follows both Rossby- and Reynolds number similarity; use of such similarity reduces computation time and allows rational comparison between different conditions. The proposed ABL model compares well with solutions of the mean momentum equations that include wind veer, if the forcing variable is employed as a free parameter.

## 1 Introduction

The interaction between the atmospheric boundary layer (ABL) and wind farms is important for wind energy, because it influences the energy yield and wind turbine life time. Many models of the ABL exist; these range from meso-scale models like the Weather Research and Forecasting (WRF) model (Skamarock et al., 2019)[1] to micro-scale models as Large-eddy simulation (LES), Reynolds-averaged Navier-Stokes (RANS), and lower fidelity models. LES is a transient method that resolves large scale turbulence, while RANS is a steady-state method that models all turbulence scales. A RANS turbulence model that can handle all relevant turbulence scales does currently not exist. Despite this fact, RANS is our method of choice because it is roughly three orders of magnitude faster compared to LES and it can be used to study trends of atmospheric wind farm flows. For example, it is possible to get good results of wind turbine wake losses in a wind farm subjected to a neutral atmospheric surface layer when using a RANS solver with modified two-equation turbulence models (van der Laan et al., 2015). In addition, RANS can simulate wind turbine interaction—meaning both wake and blockage effects—which is not trivial for engineering wind farm flow models that rely on a predefined wake (and induction) shape and wake superposition. Finally, RANS can leverage the understanding of the interaction between the ABL and wind farms, because one can add or remove components of ABL physics (representing atmospheric stability, Coriolis forces, etc.) by including or deleting the corresponding physical terms in the RANS equations.

---

[1]Notably, various 'planetary boundary layer (PBL) schemes' are available to choose from in WRF, each of which models the ABL, in a manner analogous to so-called single-column (SCM) models that are one-dimensional parameterizations of the ABL.





The use of higher fidelity ABL inflow models in RANS for wind farm flows is a research area of both practical and academic interest. One can include the effects of surface-layer atmospheric stability on a wind turbine wake using analytical profiles following Monin-Obukhov Similarity Theory ('MOST', Monin and Obukhov, 1954), as shown by e.g. Doubrawa et al. (2020), although it is not expected to be accurate for large wind turbines that operate outside the atmospheric surface layer. An idealized

model of the ABL can also be employed in RANS; for example the two-equation turbulence model of Apsley and Castro (1997) includes Coriolis forces, a constant pressure gradient, and a turbulence length-scale limiter that determines the ABL depth. Neither MOST nor ABL inflow models require a temperature equation or a buoyancy contribution in the vector momentum equation; the effect of atmospheric stability can be modeled e.g. by source terms in the turbulence model equations that only depend on velocity gradients (MOST), or via limitation of the turbulence length-scale (ABL). An advantage of MOST is that

for a prescribed (fixed) Obukhov length, the shape of the inflow profile is independent of wind speed. As a consequence, the simulated normalized wake losses in a wind farm using a MOST inflow follow Reynolds number similarity, as shown in van der Laan et al. (2020a). This is because the viscous forces can be neglected due to the high Reynolds number of atmospheric wind farm flows, and all external forces (wind turbine forces) scale as $\mathcal{U}^2/\mathcal{L}$, with $\mathcal{U}$ and $\mathcal{L}$ as characteristic velocity and length scales, respectively. The wind speed independence can be exploited when calculating the wake effects of a wind farm for

different wind speeds in a single wind farm simulation: different wind speed flow cases are run consecutively by scaling the wind turbine controller without changing the inflow profile, as shown in van der Laan et al. (2019). This method reduces the total number of iterations required to simulate multiple wind speed cases by a factor 2–3, because only local changes in the flow field need to be solved for since the global inflow is kept constant. The ABL inflow model does not follow Reynolds number similarity because the Coriolis force in the momentum equations scale linearly with $\mathcal{U}$, instead of $\mathcal{U}^2/\mathcal{L}$. However, the

ABL inflow model does follow Rossby similarity, where the ABL profiles are only dependent on two Rossby numbers if the height $z$ is normalized as $(z + z_0)f_c/G$, with $z_0$ as the roughness length, $f_c$ as the Coriolis parameter and $G$ as the geostrophic wind speed. The downside of Rossby similarity is that it cannot be used to speed up wind farm simulations as was done for MOST inflow profiles obeying Reynolds similarity, because the wind turbine size (hub height and rotor diameter) obviously does not scale by $f_c/G$.

In the this article, we present a new pressure-driven ABL model, which can be employed to follow both Reynolds and Rossby similarity. The ABL profiles of the proposed model are very similar to the ABL model of Apsley and Castro (1997) including ABL depth, by applying a momentum source term that represents the balance between Coriolis force and a fixed pressure gradient, but without turning of the wind with height (veer). Several authors (Wilson et al., 1998; Parente et al., 2011; Cindori et al., 2020) have developed unidirectional atmospheric inflow models for two- or three-dimensional simulations of complex

terrain, urban areas and forests, mainly for the purpose of wind tunnel validation where the turbulent kinetic energy varies with height. These models do not include an ABL depth, and could be interpreted as ASL models. Our proposed ABL model can be used as an inflow model for three-dimensional wind farm simulations to isolate the effects of wind veer or ABL depth, when the results are compared with wind farm simulations using an inflow based on the ABL model of Apsley and Castro (1997) (with wind veer) or a neutral surface layer, respectively. Isolating the effect of wind veer can be of interest for wake steering

control studies, where wind veer can have a significant impact as discussed by Brugger et al. (2020). In addition, it is possible





to use the model to obtain Reynolds similarity and employ it as an inflow model for wind farm simulations where wind speed flow cases are simulated consecutively to reduce the number of required iterations. The pressure-driven ABL model is not equivalent to a common pressure-driven half channel flow, because it includes a source term representing a balance between a constant pressure gradient and a type of Coriolis force, and the wind veer is removed by considering the scalar momentum

equation in the direction of the mean wind (in effect swapping the $U$ and $V$ momentum source terms and applying the same sign). While the resulting equations may not initially appear to make sense physically, the simulated ABL profiles are very similar as the ABL profiles from the ABL model of Apsley and Castro (1997) where the correct equations are employed. Furthermore, the proposed ABL model has a physical and mathematical basis, derived from a scalar momentum equation (in the mean wind direction), as shown in Section 3 (following similar ideas as seen in e.g. Sogachev et al., 2005). Section 2 first

gives an introduction of idealized ABL models in RANS, which is needed to understand the derivation of the pressure-driven ABL model. Rossby- and Reynolds number similarity of the proposed ABL model is discussed and numerically demonstrated in Sections 5 and 6, respectively. A comparison and application of the ABL models with and without veer is presented in Section 7.

## 2    Idealized ABL modeling in RANS

A steady-state idealized atmospheric boundary layer can be modeled by the incompressible RANS equations of momentum when considering homogeneous terrain and neglecting meso-scale effects:

$$\frac{DU}{Dt} = f_c(V - V_G) + \frac{d}{dz}\left(\nu_T \frac{dU}{dz}\right) = 0, \qquad \frac{DV}{Dt} = -f_c(U - U_G) + \frac{d}{dz}\left(\nu_T \frac{dV}{dz}\right) = 0, \qquad (1)$$

where $U$ and $V$ are the stream-wise and lateral horizontal velocity components, $U_G$ and $V_G$ are the geostrophic velocities which represent constant pressure gradients, $f_c$ is the Coriolis parameter dependent on latitude, $z$ is the height and $t$ is the

time. In addition, $\nu_T$ is the turbulent eddy-viscosity, which is a result from employing the linear relationship of the Reynolds-stresses and strain-rate tensor following Boussinesq (1897). The boundary conditions are $U = V = 0$ at $z = z_0$, with $z_0$ as the roughness length, and $\{U, V\} = \{U_G, V_G\}$ for $z \to \infty$. Analytic solutions of Eq. (1) exist if the turbulent eddy-viscosity $\nu_T$ is set as a constant (Ekman, 1905) or defined by a linearly increasing function with height (Ellison, 1956). Such solutions tend to not compare well with observations (e.g. Jensen et al., 1984; Hess and Garratt, 2002); this motivates the use of higher fidelity

turbulence models for the eddy-viscosity. Blackadar (1962) applied the mixing-length model of Prandtl employing a prescribed turbulence length scale $\ell$ including a maximum $\ell_{\max}$:

$$\nu_T = \ell^2 \mathcal{S}, \qquad \ell = \frac{\kappa z}{1 + \frac{\kappa z}{\ell_{\max}}}, \qquad (2)$$

where $\kappa$ is the von Kármán constant (we use $\kappa = 0.4$). In addition, $\mathcal{S}$ is the magnitude of the strain-rate tensor. For $\ell << \ell_{\max}$, the neutral surface layer solution is obtained. The parameter $\ell_{\max}$ is a proxy for ABL depth and can be used to either model

neutral or stable atmospheric conditions, as discussed by Apsley and Castro (1997). One can also model the eddy-viscosity by a two-equation turbulence model including a turbulence length scale limiter, for example the $k$-$\varepsilon$ model of Apsley and Castro



(1997):

$$\nu_T = C_\mu \frac{k^2}{\varepsilon},$$

$$\frac{Dk}{Dt} = \frac{d}{dz}\left(\frac{\nu_T}{\sigma_k}\frac{dk}{dz}\right) + \mathcal{P} - \varepsilon, \qquad \frac{D\varepsilon}{Dt} = \frac{d}{dz}\left(\frac{\nu_T}{\sigma_\varepsilon}\frac{d\varepsilon}{dz}\right) + \left(\left[C_{\varepsilon,1} + (C_{\varepsilon,2} - C_{\varepsilon,1})\frac{\ell}{\ell_{\max}}\right]\mathcal{P} - C_{\varepsilon,2}\varepsilon\right)\frac{\varepsilon}{k} \qquad (3)$$

where $C_\mu$ is a constant, $k$ is the turbulent kinetic energy and $\varepsilon$ is its dissipation. Both $k$ and $\varepsilon$ are modeled by a transport

equation, where $\mathcal{P}$ is the mechanical production of turbulence, $\ell = C_\mu^{3/4} k^{3/2}/\varepsilon$ is model based local turbulence length scale, and $C_{\varepsilon,1}, C_{\varepsilon,2}, \sigma_k, \sigma_\varepsilon$ are model constants. The two-equation turbulence model also provides results of the turbulence intensity, which is not the case for the mixing-length model. In previous work (van der Laan et al., 2020b), we have shown that the analytic solutions of Ekman (1905) and Ellison (1956) are bounds of the mixing-length model of Blackadar (1962) and the two-equation model of Apsley and Castro (1997) for $\ell_{\max} \to 0$ and $\ell_{\max} \to \infty$. In addition, the results of the numerical models

follow a Rossby similarity and all possible solutions of the ABL can be defined by two Rossby numbers based on different length scales (van der Laan et al., 2020b):

$$\mathrm{Ro}_0 \equiv \frac{G}{|f_c|z_0}, \qquad \mathrm{Ro}_\ell \equiv \frac{G}{|f_c|\ell_{\max}}. \qquad (4)$$

Here, $\mathrm{Ro}_0$ is the well known surface Rossby number and $\mathrm{Ro}_\ell$ is a Rossby number based on the maximum turbulence length scale. The Rossby similarity applies to the normalized ABL profiles, where the height $z$ is normalized as $(z + z_0)f_c/G$ and

the flow variables are normalized by $G$ and $\ell_{\max}$. The two-equation turbulence model of Apsley and Castro (1997) is only applicable to flat terrain but it can used as an inflow model for atmospheric wind farm flows in homogeneous terrain and roughness, as performed in previous work (van der Laan and Sørensen, 2017b).

## 3   A pressure-driven model of the ABL without wind veer

Our goal is to develop a pressure-driven one-dimensional model of the idealized ABL in terms of wind speed, but without wind

veer. One can derive such a model by combining the momentum equations of $U$ and $V$, i.e. Eq. (1), and rewriting them as a single equation in terms of the magnitude of geostrophic deficit (Wyngaard, 2010): $\hat{S} \equiv \sqrt{(U - U_G)^2 + (V - V_G)^2}$. Sogachev et al. (2005) also derived a momentum equation of wind speed using $S \equiv \sqrt{U^2 + V^2}$ for 2D flows, but we will use normalized velocity variables for derivation and transform the final result back to the common velocity variables. An equation for $\hat{S}$ can be derived in a number of ways. While a textbook method is to write Eq. (1) in the complex form (Wyngaard, 2010)

$$\frac{d}{dz}\left(\nu_T \frac{dW}{dz}\right) = i f_c (W - W_G) \qquad (5)$$

where $W \equiv U + iV$, $W_G \equiv U_G + iV_G$ and $i^2 = -1$, here we instead use the components in order to keep our result clear. Taking the sum of $(\hat{U} + \hat{V})DU/Dt$ and $(-\hat{U} + \hat{V})DV/Dt$ from Eq. (1), using the normalized variables $\hat{U} \equiv U - U_G$ and





$\hat{V} \equiv V - V_G$,[2] and defining the wind direction as $\hat{\varphi} = \arctan(\hat{V}/\hat{U})$, we get

$$0 = f_c\left(\hat{U}^2 + \hat{V}^2\right) + \hat{U}\frac{d}{dz}\left(\nu_T\frac{d\hat{U}}{dz}\right) + \hat{V}\frac{d}{dz}\left(\nu_T\frac{d\hat{U}}{dz}\right) - \hat{U}\frac{d}{dz}\left(\nu_T\frac{d\hat{V}}{dz}\right) + \hat{V}\frac{d}{dz}\left(\nu_T\frac{d\hat{V}}{dz}\right),$$

$$= f_c\left(\hat{U}^2 + \hat{V}^2\right) + \frac{d}{dz}\left(\nu_T\left[\hat{U}\frac{d\hat{U}}{dz} + \hat{V}\frac{d\hat{V}}{dz}\right]\right) - \nu_T\left(\left[\frac{d\hat{U}}{dz}\right]^2 + \left[\frac{d\hat{V}}{dz}\right]^2\right) - \frac{d}{dz}\left(\nu_T\left[\hat{U}\frac{d\hat{V}}{dz} - \hat{V}\frac{d\hat{U}}{dz}\right]\right),$$

$$= f_c\hat{S}^2 + \frac{d}{dz}\left(\nu_T\hat{S}^2\left[\frac{1}{\hat{S}}\frac{d\hat{S}}{dz} - \frac{d\hat{\varphi}}{dz}\right]\right) - \nu_T\hat{S}^2\left(\left[\frac{1}{\hat{S}}\frac{d\hat{S}}{dz}\right]^2 + \left[\frac{d\hat{\varphi}}{dz}\right]^2\right),$$

$$= f_c\hat{S} + \frac{d}{dz}\left(\nu_T\frac{d\hat{S}}{dz}\right) - \nu_T\hat{S}\left(\frac{d\hat{\varphi}}{dz}\right)^2 - \frac{1}{\hat{S}}\frac{d}{dz}\left(\nu_T\hat{S}^2\frac{d\hat{\varphi}}{dz}\right). \tag{6}$$

Here, we have applied the chain and product rules of differentiation, assumed a zero geostrophic shear ($dG/dz = 0$), and the following relations (Kelly and van der Laan, 2021) for the wind veer $d\hat{\varphi}/dz$ and wind shear $d\hat{S}/dz$ are employed:

$$\hat{S}^2\frac{d\hat{\varphi}}{dz} = \hat{U}\frac{d\hat{V}}{dz} - \hat{V}\frac{d\hat{U}}{dz}, \qquad \hat{S}\frac{d\hat{S}}{dz} = \hat{U}\frac{d\hat{U}}{dz} + \hat{V}\frac{d\hat{V}}{dz}, \qquad \left(\frac{d\hat{S}}{dz}\right)^2 + \left(\hat{S}\frac{d\hat{\varphi}}{dz}\right)^2 = \left(\frac{d\hat{U}}{dz}\right)^2 + \left(\frac{d\hat{V}}{dz}\right)^2 \tag{7}$$

If we take the wind veer to be much less than $(1/\hat{S})d\hat{S}/dz$ in Eq. (6) then we recover an equation for wind speed deficit $\hat{S}$, which looks identical to Eq. 5 for the wind vector (again assuming that the geostrophic shear $dG/dz$ is zero):

$$\frac{d}{dz}\left(\nu_T\frac{d\hat{S}}{dz}\right) = -f_c\hat{S}. \tag{8}$$

Neglecting veer gives $\hat{S} = |S - G|$, so then we are not really dealing with a Coriolis force, per se. In addition, solving Eq. (8) will result in a solution for $S$ where its magnitude cannot be larger than $G$ for all $z$ (which will be further motivated and the end of this section) and we can use $\hat{S} = -(S - G)$ Thus we rewrite Eq. (8) for the wind speed as

$$\frac{d}{dz}\left(\nu_T\frac{dS}{dz}\right) = f_{pg}(S - G), \tag{9}$$

where we have replaced $f_c$ by $f_{pg}$; in lieu of $f_c(S - G)$, the replacement $f_{pg}(S - G)$ represents the *magnitude* of the pressure-gradient and Coriolis effects. One needs to employ a different $f_{pg}$ than $f_c$, in order to get similar ABL profiles of wind speed, turbulence intensity and turbulence length scale, when disregarding the Coriolis-induced wind veer; this will be shown numerically in Section 7 and motivated at the end of this Section. Upon neglecting the wind veer, $DU/Dt$ and $DV/Dt$ become decoupled; then we can write the momentum equations by taking the product of Eq. (9) and $\cos(\varphi)$ or $\sin(\varphi)$, since $\varphi$ is constant ($d\varphi/dz = 0$). Preserving the relationship between magnitudes as evoked by Eqns. (8) and (9), we then have a 1-D pressure-driven ABL model:

$$\frac{d}{dz}\left(\nu_T\frac{dU}{dz}\right) = f_{pg}(U - U_G), \qquad \frac{d}{dz}\left(\nu_T\frac{dV}{dz}\right) = f_{pg}(V - V_G). \tag{10}$$

---

[2]In meteorology $\hat{U}$ and $\hat{V}$ are also known as ageostrophic velocity components, or the negative of 'geostrophic deficit'.





Equation (10) is the basis of the proposed ABL model without wind veer. It is the same as the original set of momentum equations that describe an idealized ABL, Eq. (1); however, through solving the scalar (decoupled in $x$ and $y$) equation, the source terms are swapped and have the same sign. Furthermore, one could interpret the forcing of the ABL model as a pressure gradient, hence the subscript $pg$ in $f_{pg}$. However, the ABL model does include a balance between a fixed pressure gradient and

5    a type of Coriolis force, while a standard pressure-driven ABL model does not.

With the neglect of $d\hat{\varphi}/dz$, Eq. (7) implies

$$\frac{d\hat{S}^2}{dz} = \frac{d\hat{U}^2}{dz}\left(1 + \frac{\hat{V}^2}{\hat{U}^2}\right) \qquad \text{and} \qquad \frac{d\hat{S}}{dz} = \frac{d\hat{U}}{dz}\left(1 + \frac{\hat{V}^2}{\hat{U}^2}\right)^{1/2}. \tag{11}$$

This can seen as approximation whereby the minor effect of lateral winds provides a perturbation to the streamwise gradients, when considering the full wind shear $dS/dz$ and gradient of mean kinetic energy $dS^2/dz$. Ghannam and Bou-Zeid (2020)

10    considered the effect of veer on ABL profiles, and derived an approximate model for such; the neglected terms in Eq. (6), with the above equation, can be compared to magnitudes implied by their model.

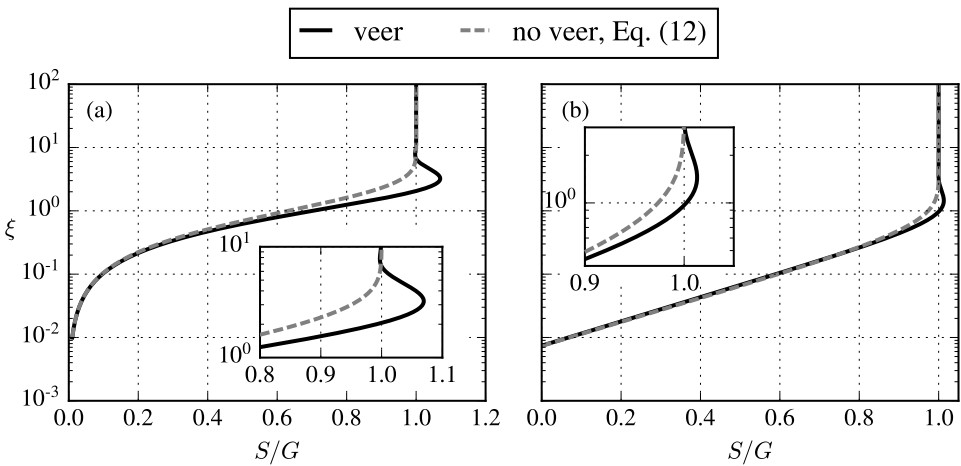

**Figure 1.** Analytic solutions with and without wind veer. **(a)** Constant $\nu_T$, $f_{pg} = |f_c|/2$. **(b)** Linear $\nu_T$, $f_{pg} = |f_c|$ and $\mathrm{Ro}_0 = 10^5$.

Analytic solutions of the wind speed profile can be derived from Eq. (10) using a constant or a linearly increasing eddy viscosity similarly to Ekman (1905) and Ellison (1956), respectively, using the original equation including wind veer, Eq. (5). The constant and linear eddy viscosity solutions of the ABL model without veer become

$$\begin{aligned} &\text{Constant } \nu_T: && S(\xi) = G\left[1 - \exp(-\xi)\right], \\ &\text{Linear } \nu_T = \kappa u_{*0} z: && S(\xi) = G\left[1 - cK_0(2\xi)\right], \quad c = 2u_{*0}/(\kappa G) = -\left[\gamma_e + \tfrac{1}{2}\ln\left(z_0 f_{pg}/(\kappa u_{*0})\right)\right]^{-1}, \end{aligned} \tag{12}$$

$\xi = z\sqrt{f_{pg}/\nu_T}$ as a normalized height, $K_0$ as the zero-order modified Bessel function of the second kind, $u_{*0}$ as the friction velocity at the surface, and $\gamma_e$ is the Euler–Mascheroni constant. The derivation of the constant $\nu_T$ solution in Eq. (12) is identical to the classical 'textbook' Ekman (1905) solution (e.g. Wyngaard, 2010), taking $i|f_c| \to f_{pg}$, which also indicates





that $\sqrt{f_{pg}} = \Re\{\sqrt{i|f_c|}\} = \sqrt{|f_c|/2}$. Note that the friction velocity in the linear $\nu_T$ solutions with and without wind veer is solved from an implicit relation of $u_{*0}/G$, through the constant $c$ derived both via $dS/dz = u_{0*}/(\kappa z)$ and $S = 0$ for $z \to z_0$. The latter is effectively a form of the geostrophic drag law, which generally arises when $G$ is used as a boundary condition in wind profile forms which include the surface stress and $z_0$ (Kelly and Troen, 2016). The solution including veer is further

discussed in van der Laan et al. (2020b), also based on Ellison (1956) and Krishna (1980). The analytical solutions, with and without wind veer are depicted in Fig. 1, which clearly shows that the wind speed of both analytic solutions without wind veer [Eq. (12)] cannot exceed the geostrophic wind speed. We also find this for the higher fidelity turbulence model closures since their solutions are bounded by the two analytic solutions (van der Laan et al., 2020b). The ABL model with wind veer includes the supergeostrophic wind speed (jet), which typically occurs below the ABL top, as predicted by the Ekman

equations (Blackadar, 1957). The jet is a consequence of the Coriolis-induced interaction of alternate horizontal momentum and stress components; it does not exist in an idealized ABL model when the wind veer—and more importantly the oscillating part of the solution (which results from the coupling of the $u$ and $v$ equations)—is removed. This is not a new insight because one could deduce it from a text book (e.g. Ch.10 of Wyngaard, 2010); however, we have shown the relation between the Coriolis-induced wind veer and the jet more explicitly by deriving two analytic solutions without wind veer [Eq. (12)], as

depicted in Fig. 1.

## 4 Methodology of numerical simulations

The methodology of the numerical one-dimensional simulations of the present article is very similar as performed in van der Laan et al. (2020b), a brief summary is presented here. The RANS simulations are carried out with a one-dimensional version of EllipSys (van der Laan and Sørensen, 2017a), which is an in-house incompressible finite volume flow solver initially developed

by Michelsen (1992) and Sørensen (1994). The numerical grid represents a $10^5$ m line with 384 cells that increase with height using an expansion ratio of 1.2 and a first cell height of $10^{-2}$ m. The number of cells is a conservative choice based on a grid refinement study, as performed in previous work (van der Laan et al., 2020b). The bottom and top boundary conditions are set as a rough wall (Sørensen et al., 2007) and symmetry boundaries, respectively. Ambient source terms in transport equations of $k$-$\varepsilon$ model are employed in order to prevent zero values (van der Laan et al., 2020b). The RANS simulations are solved

transient with a fixed large time step set to $1/|f_c|$ or $1/f_{pg}$ s and converge to a steady-state solution. The following turbulence model constants are employed: $(C_\mu, C_{\varepsilon,1}, C_{\varepsilon,2}, \sigma_k, \sigma_\varepsilon, \kappa) = (0.03, 1.21, 1.92, 1.0, 1.3, 0.4)$.

## 5 Rossby number similarity

The ABL model including wind veer follows a Rossby similarity as shown in previous work (van der Laan et al., 2020b). As a consequence, all possible normalized solutions of the ABL are only dependent on two Rossby numbers, each with a different

length scale (see Eq. 4). The proposed ABL model without wind veer, as derived in Section 3, also follows a Rossby similarity.





| $\widetilde{\mathrm{Ro}}_0, \widetilde{\mathrm{Ro}}_\ell$ | $f_{pg}[\mathrm{s}^{-1}], G[\mathrm{ms}^{-1}]$ | $f_{pg}[\mathrm{s}^{-1}], G[\mathrm{ms}^{-1}]$ | $f_{pg}[\mathrm{s}^{-1}], G[\mathrm{ms}^{-1}]$ | $f_{pg}[\mathrm{s}^{-1}], G[\mathrm{ms}^{-1}]$ |
|---|---|---|---|---|
| $10^6, 10^3$: | $5 \times 10^{-5}, 10$ | $5 \times 10^{-5}, 20$ | $10^{-4}, 10$ | $10^{-4}, 20$ |
| $10^6, 10^5$: | $5 \times 10^{-5}, 10$ | $5 \times 10^{-5}, 20$ | $10^{-4}, 10$ | $10^{-4}, 20$ |
| $10^9, 10^3$: | $5 \times 10^{-5}, 10$ | $5 \times 10^{-5}, 20$ | $10^{-4}, 10$ | $10^{-4}, 20$ |
| $10^9, 10^5$: | $5 \times 10^{-5}, 10$ | $5 \times 10^{-5}, 20$ | $10^{-4}, 10$ | $10^{-4}, 20$ |

**Figure 2.** Rossby number similarity of the proposed ABL model without wind veer for two turbulence closures. **(a-c)** Mixing-length model. **(d-g)** $k$-$\varepsilon$ model.

One can show this by writing the equation of wind speed, Eq. (9), in non-dimensional form

$$\widetilde{\mathrm{Ro}}_0 \frac{d}{dz'}\left(\left[\frac{\kappa z'}{1 + \kappa z' \widetilde{\mathrm{Ro}}_\ell/\widetilde{\mathrm{Ro}}_0}\right]^2 \left(\frac{dS'}{dz'}\right)^2\right) = S' - 1, \tag{13}$$

where $z' = z/z_0$, $S' = S/G$, and $\widetilde{\mathrm{Ro}}_\ell/\widetilde{\mathrm{Ro}}_0 = z_0/\ell_{\max}$. Here we have used the mixing-length turbulence model and prescribed the turbulence length scale of Blackadar (1962) [Eq. (2)], but the same Rossby similarity applies to the two-equation turbulence model of Apsley and Castro (1997) [Eq. (3)]. In addition, the two Rossby numbers are defined as:

$$\widetilde{\mathrm{Ro}}_0 \equiv \frac{G}{f_{pg} z_0}, \qquad \widetilde{\mathrm{Ro}}_\ell \equiv \frac{G}{f_{pg} \ell_{\max}}. \tag{14}$$



A numerical proof of the Rossby similarity of the ABL model without wind veer for both the mixing-length and two-equation turbulence models is depicted in Fig. 2. Four sets of Rossby numbers are used, and each set is simulated by four cases using two different values of $f_{pg}$ and $G$. Normalized results of wind speed, wind direction, eddy viscosity and turbulence intensity collapse and are only dependent on the two Rossby numbers.

## 6 Reynolds number similarity

If all external forces in the momentum equations scale by $\mathcal{U}^2/\mathcal{L}$, with $\mathcal{U}$ and $\mathcal{L}$ as characteristic velocity and length scales, respectively, then one can obtain Reynolds number similarity. The Reynolds number can be defined as:

$$\mathrm{Re} \equiv \frac{\mathcal{U}\mathcal{L}}{\nu} = \frac{f_c \mathcal{L}^2}{\nu}\mathrm{Ro}, \tag{15}$$

with $\nu$ as the molecular viscosity and it can be related to the Rossby number $\mathrm{Ro} \equiv \mathcal{U}/(f_c\mathcal{L})$. Wind turbine wake simulations where only pressure rotor forces are considered (and viscous rotor forces are neglected) follow Reynolds number similarity if the inflow model does as well (van der Laan et al., 2020a). For high Reynolds numbers, the wind turbine wake simulations then become independent of inflow wind speed and wind turbine size (rotor diameter $D$ and hub height $z_H$, as long $D/z_H$ is kept constant), which can be employed to reduce the total number of required iterations of parametric studies or annual energy calculations of wind farms using RANS (van der Laan et al., 2019). The Coriolis force in the ABL model with wind veer scales by $\mathcal{U}$, which means that Reynolds number similarity cannot be obtained. However, the $f_{pg}$ parameter in the pressure-driven ABL model can be redefined to obtain Reynolds number similarity:

$$f_{pg} \equiv C\frac{\mathcal{U}}{\mathcal{L}} = \frac{1}{\widetilde{\mathrm{Ro}}_0}\frac{G}{z_0}, \tag{16}$$

with $C$ as a constant, which is equivalent to setting $\widetilde{\mathrm{Ro}}_0 = 1/C$ and $\widetilde{\mathrm{Ro}}_\ell = z_0/\ell_{\max}/C$. We can allow ourselves to redefine $f_{pg}$ because it does not directly represent a physical Coriolis parameter as $f_c$ does in the ABL model with wind veer. Substitution of Eq. (16) and employing the mixing-length model of Eq. (2) in Eq. (9) leads to same results as given in Eq. (13). For a constant $\widetilde{\mathrm{Ro}}_0$, the only parameter that changes the ABL profile shape is $z_0/\ell_{\max}$, which is a proxy for normalized ABL depth. Hence, the ABL model becomes wind speed independent and one can use it as an inflow for Reynolds number independent wind turbine wake simulations for a fixed $z_0/\ell_{\max}$, which represents a fixed ABL depth or a fixed atmospheric stability. The Reynolds number independence is shown in Fig. 3 for two turbulence closures. Two values of $\widetilde{\mathrm{Ro}}_0$ are used, representing onshore and offshore roughness for latitudes around $\pm45°$. Each Rossby number is simulated with two different values of $z_0/\ell_{\max}$. Note that we have chosen to use a different set of $z_0/\ell_{\max}$ for each Rossby number because the range of meaningful $z_0/\ell_{\max}$ is dependent on the choice of $\widetilde{\mathrm{Ro}}_0$. The four resulting ABL cases are then simulated with two geostrophic wind speeds and roughness lengths. Figure 3 shows that the normalized ABL profiles are independent of $G$, as long as $\widetilde{\mathrm{Ro}}_0$ and $z_0/\ell_{\max}$ are kept constant. Figure 3 can also be interpreted as Rossby number similarity as depicted Fig. 2; however, the difference is that $f_{pg}$ in Fig. 3 is now used to keep $\widetilde{\mathrm{Ro}}_0$ constant for different values of $G$ and $z_0$.



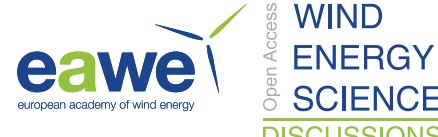

| $\widetilde{\mathrm{Ro}}_0, z_0/\ell_{\max}$ | $z_0[\mathrm{m}], G[\mathrm{ms}^{-1}]$ | $z_0[\mathrm{m}], G[\mathrm{ms}^{-1}]$ | $z_0[\mathrm{m}], G[\mathrm{ms}^{-1}]$ | $z_0[\mathrm{m}], G[\mathrm{ms}^{-1}]$ |
|---|---|---|---|---|
| $10^6, 10^{-1}$: | $10^{-1}, 10$ | $10^{-2}, 10$ | $10^{-1}, 20$ | $10^{-2}, 20$ |
| $10^6, 10^{-3}$: | $10^{-1}, 10$ | $10^{-2}, 10$ | $10^{-1}, 20$ | $10^{-2}, 20$ |
| $10^9, 10^{-4}$: | $10^{-3}, 10$ | $10^{-4}, 10$ | $10^{-3}, 20$ | $10^{-4}, 20$ |
| $10^9, 10^{-6}$: | $10^{-3}, 10$ | $10^{-4}, 10$ | $10^{-3}, 20$ | $10^{-4}, 20$ |

**Figure 3.** Reynolds number similarity of the proposed ABL model without wind veer for two turbulence closures. **(a-c)** Mixing-length model. **(d-g)** $k$-$\varepsilon$ model.

## 7  Comparison of ABL models and application to inflow profiles

In this section, one-dimensional RANS simulations are performed to compare the proposed ABL model without wind veer to the ABL model including wind veer, and we investigate the application to use the model as an inflow model. Three-dimensional RANS simulations are not performed in this article and will be carried out in future work. In addition, the $k$-$\varepsilon$ ABL model

5 of Apsley and Castro (1997) is used because it also provides an estimate of the turbulence intensity, while the mixing-length model of Blackadar (1962) does not.

Figure 4 shows a comparison between the original $k$-$\varepsilon$ ABL model with wind veer and the proposed $k$-$\varepsilon$ ABL without wind veer. Results of a stable ($\ell_{\max} = 5$ m) and a neutral ABL ($\ell_{\max} = 30$ m) are depicted for an offshore roughness length of



WIND
ENERGY
SCIENCE
DISCUSSIONS

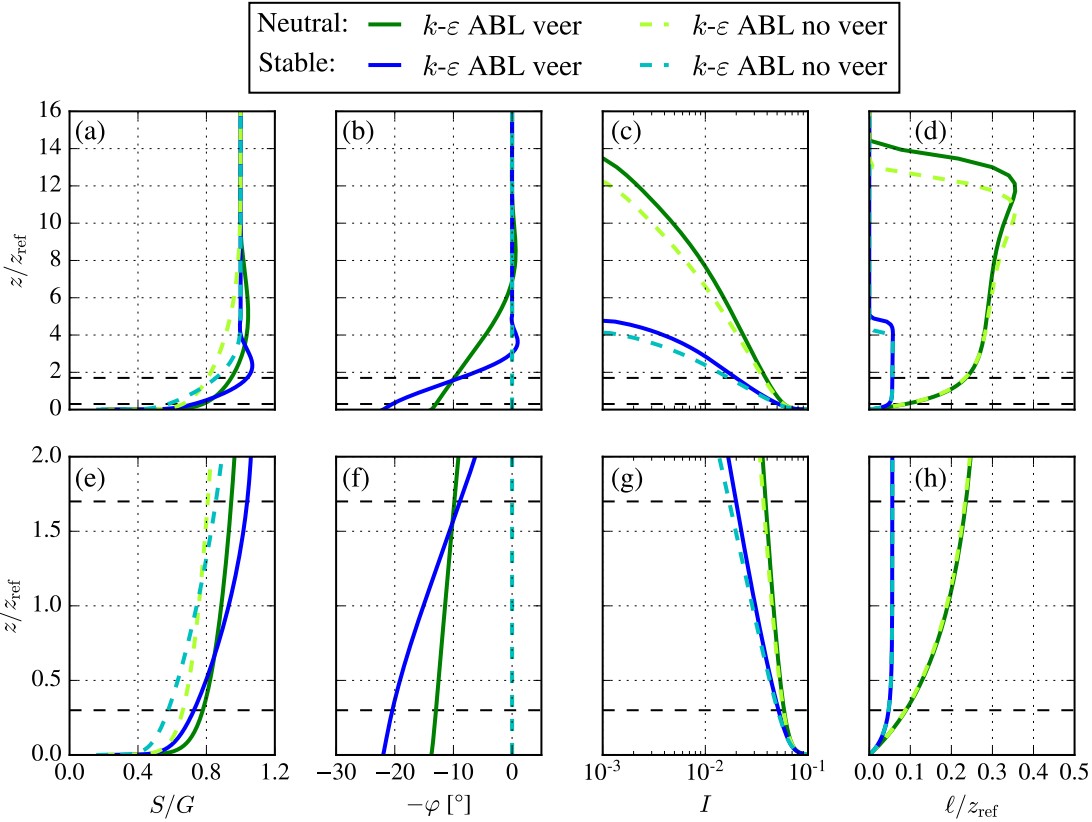

**Figure 4.** Comparison of $k$-$\varepsilon$ ABL models (with and without wind veer) for stable ($\ell_{\max} = 5$ m) and neutral ($\ell_{\max} = 30$ m) conditions offshore using $G = 10$ ms$^{-1}$, $f_c = 2f_{pg} = 10^{-4}$ s$^{-1}$ and $z_0 = 10^{-4}$ m. Bottom plots **(e-h)** are a zoomed view of the top plots **(a-d)**, focused around a 5 MW wind turbine rotor area, which is depicted as black dashed lines. **(a, e)** Wind speed. **(b, f)** Wind direction. **(c, g)** Turbulence intensity. **(d, h)** Turbulence length scale.

$10^{-4}$ m, a geostrophic wind speed of 10 ms$^{-1}$ and a Coriolis parameter of $10^{-4}$ s$^{-1}$. In addition, the parameter $f_{pg}$ of the ABL model without veer is set as $f_{pg} = |f_c|/2$, as suggested by the classic Ekman (1905) solution discussed in Section 3. The ABL profiles of Fig 4 could be used as inflow profiles for offshore wind farm simulations, and we have chosen to normalize the results by a reference height, $z_{\mathrm{ref}} = 90$ m, which corresponds to the hub height of the NREL-5MW reference wind turbine
5    (Jonkman et al., 2009). In addition, the swept rotor area is marked as dashed black lines in Fig. 4 and the bottom plots are zoomed views of the top plots, with a focus on the ABL around the fictitious wind turbine. Figure 4a) shows the wind speed and it is clear that the supergeostrophic jet (Blackadar, 1957) is a feature that cannot be predicted by the ABL model without wind veer, as also found for the analytic solutions from Eq. (12). As a consequence, the wind shear in the pressure-driven ABL model is smaller than the original ABL model including wind veer, as seen in Fig. 4e). Figures 4b) and f) show that
10    the proposed ABL model predicts a zero wind veer as intended. Finally, the turbulence intensity and length scale are similar



between the ABL models around the wind turbine rotor area, but are different for higher altitudes, as shown in Figs. 4c) and d).

It is possible to use $f_{pg}$ as a free parameter in the pressure-driven ABL model, for a given height, to give results approximating those from the original ABL model considering veer around a reference height. One can motivate this choice, because

$f_{pg}$ does not simply represent the Coriolis parameter $f_c$; while it is a proxy for $f_c$ with use of the scalar wind speed equation, it also contains the effects of the neglected wind veer, since we can write

$$f_{pg} = f_c - \nu_T \left( \frac{d\hat{\varphi}}{dz} \right)^2 - \frac{1}{\hat{S}^2} \frac{d}{dz} \left( \nu_T \hat{S}^2 \frac{d\hat{\varphi}}{dz} \right). \tag{17}$$

Here, we have set Eq. (9) equal to the final result of Eq. (6). It is not trivial to solve for Eq. (17) and we choose to obtain $f_{pg}$ from a library of pre-calculated ABL profiles, which are only dependent on the two Rossby numbers $\widetilde{Ro}_0$ and $\widetilde{Ro}_\ell$. For wind

farm simulations, one would like to obtain an inlet ABL profile for a desired reference wind speed $S_{ref}$ and turbulence intensity $I_{ref}$, specified at a reference height $z_{ref}$ for a given site where $z_0$ and $f_c$ are known. In Appendix A, a procedure is presented how to obtain a desired ABL profile using pre-calculated libraries of normalized ABL profiles based on Rossby number similarity. Examples of neutral and stable ABL profiles are made by using the chosen and derived values listed in Table 1, and the results of both ABL models are depicted in Fig. 5. Here, we have used $f_{pg}$ as a free parameter and the geostrophic wind

speed is derived differently for both ABL models. In general, we find that $f_{pg} < f_c$ (as discussed previously) and $G_{pg} > G$. A higher geostrophic wind speed is required in the ABL model without wind veer to compensate for a reduced wind shear at $z = z_{ref}$ caused by the lack of the supergeostrophic jet. Figure 5 shows that a close match between the ABL models can be achieved in terms of wind speed, turbulence intensity and turbulence length scale, especially around the rotor area. Hence, the pressure-driven ABL model can be used as in inflow model in wind farm simulations to isolate the effect of wind veer when it

is compared to an inflow model based on the original ABL model including wind veer. The main differences between the ABL model can be found in terms of wind speed near the supergeostrophic jet and above. For very stable conditions, the location of the jet could approach the wind turbine rotor and the effect of wind veer cannot be isolated from the effect of the jet unless one considers the jet to be part of the effect of wind veer as motivated previously.

| Case | Input parameters | | | | | Derived parameters | | | |
|---|---|---|---|---|---|---|---|---|---|
| | $I_{ref}$ | $S_{ref}$ [ms$^{-1}$] | $z_{ref}$ [m] | $z_0$ [m] | $f_c$ [s$^{-1}$] | $G$ [ms$^{-1}$] | $\ell_{max}$ [m] | $f_{pg}$ [s$^{-1}$] | $G_{pg}$ [ms$^{-1}$] |
| Neutral | 0.045 | 8 | 90 | $10^{-4}$ | $10^{-4}$ | 8.92 | 22.3 | $4.37 \times 10^{-5}$ | 11.0 |
| Stable | 0.03 | 8 | 90 | $10^{-4}$ | $10^{-4}$ | 8.42 | 5.01 | $4.36 \times 10^{-5}$ | 11.3 |

**Table 1.** Summary of input and derived parameters for ABL models.

In previous work, the $k$-$\varepsilon$ ABL model with wind veer has been applied as an inflow to wind farm simulations using RANS to

investigate the effect of Coriolis forces (van der Laan and Sørensen, 2017b) and the $k$-$\varepsilon$ ABL was coupled with a $k$-$\varepsilon$ developed for wake simulations under neutral ASL conditions. A similar coupling could be made with the proposed ABL model without wind veer, which will be investigated in future work.



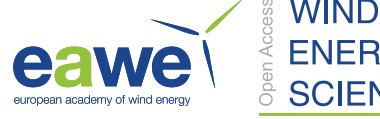

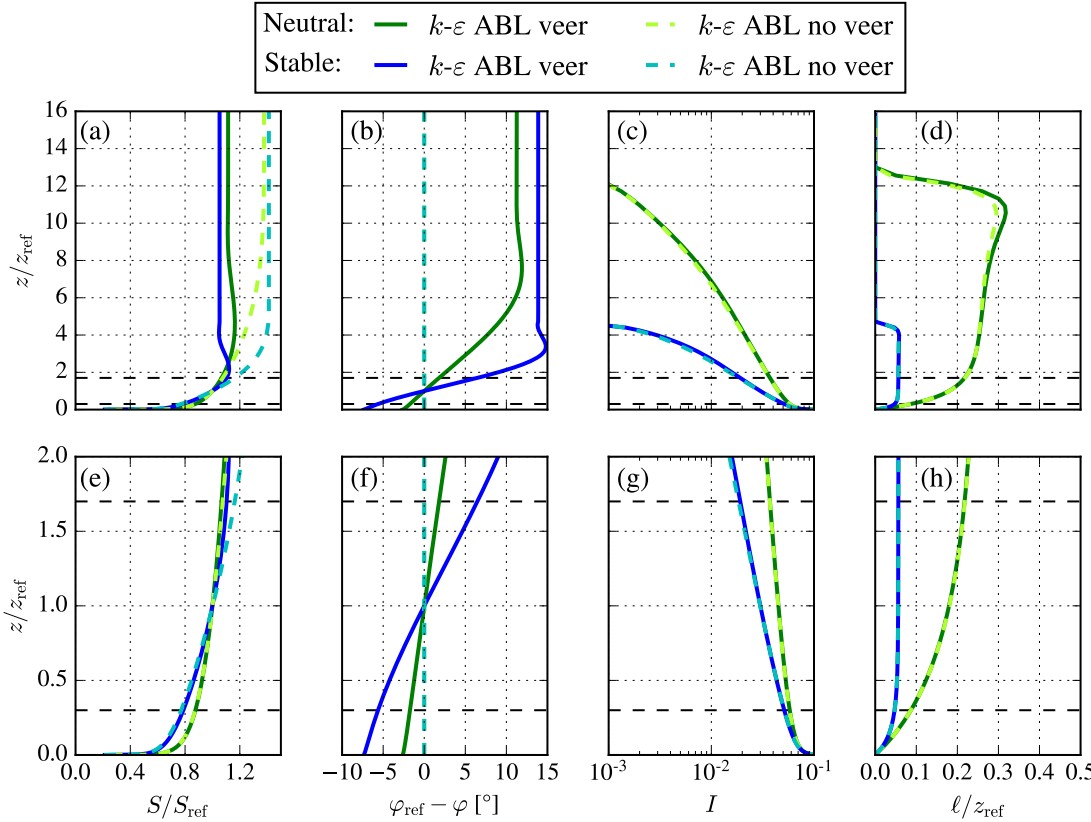

**Figure 5.** Comparison of $k$-$\varepsilon$ ABL models (with and without wind veer) for stable and neutral conditions offshore using $f_c \neq 2f_{pg}$ and parameters from Table 1. Bottom plots **(e-h)** are a zoomed view of the top plots **(a-d)**, focused around a 5 MW wind turbine rotor area, which is depicted as black dashed lines. **(a, e)** Wind speed. **(b, f)** Wind direction. **(c, g)** Turbulence intensity. **(d, h)** Turbulence length scale.

It should be noted that both $k$-$\varepsilon$ ABL models (with and without wind veer) cannot be used as an inflow model for complex terrain simulations in RANS because the employed global length scale limiter of Apsley and Castro (1997) does not perform well when the turbulence length scale associated to the terrain is larger than $\ell_{\max}$. We plan to modify the length scale limiter in future work to overcome this issue. However, the proposed momentum source from Eq. (10) can still be used in combination
5   with an alternative turbulence model suited for complex terrain.

## 8   Conclusions

We have proposed a pressure-driven model of the mean ABL without wind veer, based on the streamwise (scalar) momentum equation. One-dimensional RANS simulations of the pressure-driven ABL model are performed to show that the model follows both Rossby- and Reynolds number similarity. The similarities can be employed to quickly find a desired ABL profile based





on a pre-calculated library of ABL profiles, which can be used as an inflow profile for three-dimensional RANS simulations of wind farms. The pressure-driven ABL model compares well with an ABL model including wind veer, if the forcing variable $f_{pg}$ is used a free parameter. The largest differences between the models is found near the location of the super-geopstrophic jet and above (i.e., around the top of the ABL), because the pressure-driven ABL model cannot represent wind speeds exceeding the

geostrophic wind. The absence of the geostrophic jet in the pressure-driven ABL model is related to the lack of Coriolis-induced wind veer (lack of coupling between the equations for $DU/Dt$ and $DV/Dt$), as explicitly shown by analytic solutions without wind veer for constant and linearly increasing eddy viscosity. The difference between the ABL models can become important for shallow boundary layers representing very stable atmospheric conditions, if the pressure-driven ABL model is used as an inflow profile for three-dimensional RANS simulations of large wind turbines. Despite this challenge, one can employ the

pressure-driven ABL model to isolate the effect of wind veer or ABL depth, when it is compared to an ABL model including wind veer or an ASL model, respectively. In addition, the Reynolds number similarity of the pressure-driven ABL model can be used to perform parametric studies of the effect of wind speed on wind farm flow simulations more quickly, similar to using an ASL inflow model (van der Laan et al., 2019). The proposed ABL model in combination with the turbulence model of Apsley and Castro (1997) cannot yet be used for complex terrain simulations, since the length-scale limiter does not behave

appropriately over such terrain. However, it is possible to use the momentum source term of the ABL model in combination with a turbulence model suited for complex terrain; such work is still under development. Further, 'softening' of the effect of the length-scale limiter, to capture the influence of the strength of ABL-capping inversion (Kelly et al., 2019b), is also needed (and underway) to better capture the top-down effects entraining momentum into the windfarm (e.g. Kelly et al., 2019a).

*Code and data availability.*  The numerical results are generated with DTU's proprietary software, although the data presented can be made
available by contacting the corresponding author.

*Author contributions.*  MPVDL has performed the simulations, proposed the pressure-driven ABL model, obtained the model-based Rossby and Reynolds number similarity and produced all figures. All authors have contributed to the derivation and development of the pressure-driven ABL model, and article writing.

*Competing interests.*  The authors declare that they have no conflict of interest.





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



**Appendix A: Obtaining a desired ABL profile using Rossby number similarity**

The results of the $k$-$\varepsilon$ ABL model of Apsley and Castro (1997) (with wind veer) and the proposed ABL model without wind veer from Section 3 can both be used as inflow profiles for three-dimensional wind farm simulations for homogeneous terrain. Wind farm simulations are often run as flow cases, where a desired turbulence intensity $I_{\mathrm{ref}}$ and wind speed $S_{\mathrm{ref}}$ is set at a

reference height $z_{\mathrm{ref}}$. In this section, a methodology is presented on how to use the Rossby number similarity (Section 5) to quickly find the geostrophic wind speed and maximum turbulence length scale that corresponds to the desired inflow profile from a pre-calculated library of all possible normalized ABL profiles based on two Rossby numbers [Eq. (4) or Eq. (14)]. In order to find a unidirectional ABL profile that matches an ABL profile with veer we perform the following steps:

1. Simulate non-dimensional libraries of ABL profiles for both models (with and without wind veer) based on the two

Rossby numbers (i.e. both $\widetilde{\mathrm{Ro}}_0$ and $\widetilde{\mathrm{Ro}}_\ell$ as well as $\mathrm{Ro}_0$ and $\mathrm{Ro}_\ell$). These Rossby numbers can be made of any combination of $G$, $f_c$, $z_0$ and $\ell_{\max}$ as shown in Figure 2. For example, choose $G = G_{\mathrm{lib}} = 10$ m/s, $f_{pg} = f_c = f_{c,\mathrm{lib}} = 10^{-4}$ s$^{-1}$, $\ell_{\max,\mathrm{lib}} = G_{lib}/(f_{c,\mathrm{lib}}\mathrm{Ro}_\ell)$ and $z_0 = G_{\mathrm{lib}}/(f_{c,\mathrm{lib}}\mathrm{Ro}_0)$. We have chosen to use a parametric study of two Rossby numbers in logarithmic space, e.g. $(\mathrm{Ro}_0, \mathrm{Ro}_\ell) = (\widetilde{\mathrm{Ro}}_0, \widetilde{\mathrm{Ro}}_\ell) = (10^a, 10^b)$ with $a = [5, 10]$ using a spacing of 0.2 and $b = [2, 4.5]$ with a spacing of 0.1 for $b < 3.5$ and a finer spacing of 0.05 for $b > 3.5$. The results of are stored as function of a

normalized height $z_{\mathrm{norm}} = (z + z_0)f_{c,\mathrm{lib}}/G_{\mathrm{lib}}$.

2. Set $z_{0,\mathrm{ref}}$, $f_{c,\mathrm{ref}}$, $S_{\mathrm{ref}}$, $I_{\mathrm{ref}}$ and $z_{\mathrm{ref}}$.

3. Find $\mathrm{Ro}_0$ and $\mathrm{Ro}_\ell$ from the ABL library with veer that satisfy the reference values of Step 2 and calculate $G$ and $\ell_{\max}$:

   (a) For each $(\mathrm{Ro}_0,\mathrm{Ro}_\ell)$-pair interpolate $z_{\mathrm{norm}}$ where $I = I_{\mathrm{ref}}$ is obtained and calculate the corresponding geostrophic wind speed $G(\mathrm{Ro}_0, \mathrm{Ro}_\ell) = (z_{\mathrm{ref}} + z_{0,\mathrm{ref}})f_{c,\mathrm{ref}}/z_{\mathrm{norm}}$ and wind speed $S(\mathrm{Ro}_0, \mathrm{Ro}_\ell)$.

(b) Curve $A$ in Figure A1 is a set of points satisfying $(\{\mathrm{Ro}_{\ell,A}\}, \{\mathrm{Ro}_{0,A}\}) = \{(\mathrm{Ro}_\ell, \mathrm{Ro}_0)|S(\mathrm{Ro}_\ell, \mathrm{Ro}_0) = S_{ref}\}$.

   (c) Curve $B$ in Figure A1 represents $(\{\mathrm{Ro}_{l,B}\}, \{\mathrm{Ro}_{0,B}\}) = (\{\mathrm{Ro}_{l,A}\}, \{G_A\}/(f_{c,\mathrm{ref}}z_{0,\mathrm{ref}}))$, where $G_A$ corresponds to extracted $G$ values from curve $A$.

   (d) $\mathrm{Ro}_0$ and $\mathrm{Ro}_\ell$ can be obtained by the intersection of curves $A$ and $B$.

   (e) Calculate $\ell_{\max} = z_{0,\mathrm{ref}}\mathrm{Ro}_0/\mathrm{Ro}_\ell$ and $G = z_{0,\mathrm{ref}}f_{c,\mathrm{ref}}\mathrm{Ro}_0$.

4. Find $\widetilde{\mathrm{Ro}}_0$ and $\widetilde{\mathrm{Ro}}_\ell$ from the ABL library without wind veer similar to steps 3a-d and calculate $f_{pg} = f_{c,\mathrm{ref}}\mathrm{Ro}_0/\widetilde{\mathrm{Ro}}_0$, then correct the geostrophic wind speed as $G_{pg} = GS_{\mathrm{ref}}/S_{pg}$ using Reynolds number similarity.

Here, the subscript $pg$ is used for the parameters corresponding to the ABL model without wind veer, where $S_{pg}$ is the obtained wind speed at $z_{\mathrm{ref}}$ before correcting the geostrophic wind speed $G$ to $G_{pg}$. In addition, we use $\mathrm{Ro}_0$ and $\mathrm{Ro}_\ell$ in logarithmic space in steps 3a-d. If the ABL libraries only contain a few profiles for different Rossby numbers, then obtaining the desired

Rossby numbers may lead to errors. In case this one could use the ABL libraries as an initial guess for a numerical optimization

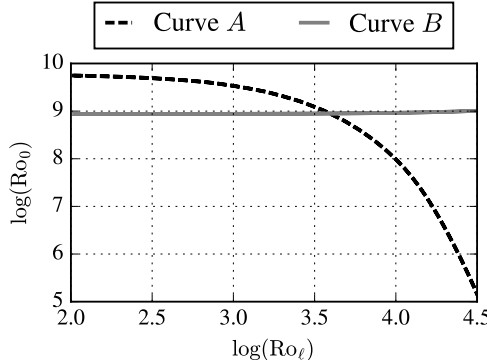

**Figure A1.** Example of obtaining a set of Rossby numbers from an ABL library with wind veer, Step 3d.

that results in the required $G$ and $\ell_{\max}$ and $f_{pg}$ and $G_{pg}$. An example of obtaining a set of Rossby number from an ABL library with wind veer (Step 3c) is depicted in Fig. A1. The example represents the neutral case as listed in Table 1.