# Peer review of "A pressure-driven atmospheric boundary layer model satisfying Rossby- and Reynolds number similarity"

_Wind Energy Science, 2020_

## Author Comment (AC1)

**Reply to reviewers**

**March 9, 2021**

We would like to thank the two reviewers for their detailed feedback and suggestions to improve the article. In the next sections, the reviewers comments are copied and answered per comment (blue color). An additional document is provided that highlights all modifications with respect to the initial submitted version.

**Reviewer 1 (Javier Sanz Rodrigo)**

Elegant formulation of an ABL model with numerical properties to allow efficient computation of wind conditions by dropping dependencies on the wind sheer. The model is motivated by previous papers from the authors grounded on the concept of Re and Ro similarity and provide a practical approach to the parameterization of the model using pre- computed ABL simulations. I have only minor remarks and suggestions that can help in the understanding the derivation of the model. In particular it is important to mention how stability effects are introduced without using a potential temperature equation.

1. Abstract: "for isolating the effects..." this part does not read well, please reformulate.

We have reformulated: for isolating the effects of wind veer and ABL depth to to separately demonstrate the impact of wind veer and ABL depth.

2. P1.19: Please add an example of a reference for RANS dealing with wake and blockage.

We referred to previous work (van der Laan et al., 2015) (which was referenced earlier in the introduction) and added a new reference to Bleeg et al. (2018).

3. P1.22: Please add an example of a reference for RANS switching physics components on and off.

This is a topic that we are addressing in detail in an submitted conference article (Wake Conference 2021) We have added a bit of context explaining the concept of switching physics components on and off: For example, a RANS model can be used to model the effect of non-neutral atmospheric stability (van der Laan et al., 2020a). When the model operates in neutral mode, all model components that represent non-neutral conditions are switched off.

4. P2.23: avoid using "obviously" when stating any modeling hypothesis. "It ain't what you don't know that gets you into trouble. It's what you know for sure that just ain't so." - Mark Twain (one of my favorite quotes that is very applicable to science).

We fully agree that the word obviously should be used with care in research, we have removed it. In addition, we have added that the height can also be scaled as  $(z + z_0)/z_0$  to obtain Rossby similarity since it holds when the surface layer Rossby number,  $\text{Ro}_0 = G/(f_c z_0)$ , is constant.

5. P3.15: The ABL formulation does not include an equation for potential temperature so there is an implicit assumption that the equations are only applicable in neutral conditions. However, later on you show results for stable conditions which, in practice, are dependent on the value of lmax. I think that

when you start the description of the model in chapter 2 you should make it clear how you are dealing with stability so that the reader knows that the model is more generally applicable that just neutral conditions. This is particularly confusing when you define lmax based on Blackadar's equation (2) which was meant to be used in neutral conditions. If I understood correctly, as described in the annex and illustrated in Table 1, lmax and other forcings are derived based on best-fit between the desired reference wind conditions and a pre-simulated library of ABL profiles. This is a practical approach to defining these quantities that are not typically measured in real word campaigns. Instead they become tuning parameters of the model to match the desired profile.

It is correct that a potential temperature is not employed. This is because we would like to obtain both Reynolds- and Rossby number similarity. In the introduction of the submitted article, we had mentioned the following: Neither MOST nor ABL inflow models require a temperature equation or a buoyancy contribution in the vector momentum equation; the effect of atmospheric stability can be modeled e.g. by source terms in the turbulence model equations that only depend on velocity gradients (MOST), or via limitation of the turbulence length-scale (ABL). We agree that we can remind the reader again about this in Section 2, where we have added the following after eq.(2): For  $\ell_{\max} \to \infty$ , neutral conditions are obtained that correspond to the analytic solution of Ellison (1956), as discussed in van der Laan et al. (2020b). For small values of  $\ell_{\max}$ , an ABL profile is obtained that has the characteristics of stable conditions; a shallow ABL, a strong shear and wind veer, and a small eddy viscosity, all with respect to neutral conditions. Hence, a potential temperature equation is not necessary and the effects of a stable ABL are solely modeled by a limitation of the turbulence length scale.

The calculated values of G,  $\ell_{\max}$  and  $f_{pg}$  are indeed obtained from pre-calculated ABL libraries employing Rossby similarity. A turbulence intensity and wind speed at a reference height are the input parameters that can be based on atmospheric measurements. In addition, one can choose a roughness length and a Coriolis parameter to represent a given site (although the choice in roughness length will set the maximum turbulence intensity that one can obtain). Note that this was mentioned in the submitted article, just under eq.(17): For wind farm simulations, one would like to obtain an inlet ABL profile for a desired reference wind speed  $S_{ref}$  and turbulence intensity  $I_{ref}$ , specified at a reference height  $z_{ref}$  for a given site where  $z_0$  and  $f_c$  are known. In Appendix A, a procedure is presented of how to obtain a desired ABL profile using pre-calculated libraries of normalized ABL profiles based on Rossby number similarity.

6. P4.16: "it can be used"

Adopted.

7. P5.Eq.(8): "If we take the wind veer to be much less than  $(1/\hat{S})d\hat{S}/dz$ . It is not straightforward to understand if this assumption holds in general. I guess that the net effect of this assumption is apparent in Figure 1 but you might as well plot the wind veer components vs the wind shear components of eq. (6) to show their relative importance with height and then the approximation will be better substantiated.

Thanks for raising this point. We have added a clarification under Eq. (8): The assumption  $d\hat{\varphi}/dz \ll (1/\hat{S})d\hat{S}/dz$  could be considered a weak assumption since it in principle allows for some veer. However, for some cases this is violated; e.g. in the Ekman (1905) solution (constant eddy viscosity) both terms are equal. A stronger and simpler assumption, which also leads to Eq. (8), is to simply take  $d\hat{\varphi}/dz = 0$  in Eq. (6).

We agree that a plot of  $d\hat{\varphi}/dz$  and  $(1/\hat{S})d\hat{S}/dz$  could be interesting for complex profiles of the eddy viscosity; however, we prefer to it leave it out because we are in the process of writing another dedicated article about wind shear and wind veer. In addition, one can already see the effect of assuming a zero wind veer on the wind speed profile in Fig. 1, as the reviewer pointed out as well.

8. P6,8: "This can be seen as an approximation"

Adopted.

9. P7,26: For completeness, can you specify where this set of constants is coming from?

We have added a reference to Sørensen (1994).

10. P8,Figure 2: Please define the symbol "I" of subplot (g) in the text or in the caption.

We have added the definition of the turbulence intensity in the text, at the end of Section 5.

11. P9,21: Can you elaborate why z0/lmax is a proxy for normalized ABL depth? Furthermore, why does it "represent a fixed ABL depth or a fixed atmospheric stability?

We have removed which is a proxy for a normalized ABL depth and discussed it later after introducing Fig.3, at the end of Section 3: In addition, it is clear that  $z_0/\ell_{\text{max}}$  is proxy for a normalized (reciprocal) ABL depth, when comparing the profiles in pairs for a constant  $\widetilde{\text{Ro}}_0$  ( $\widetilde{\text{Ro}}_0 = 10^6, z_0/\ell_{\text{max}} = 10^{-1}$  compared to  $\widetilde{\text{Ro}}_0 = 10^6, z_0/\ell_{\text{max}} = 10^{-3}$  and  $\widetilde{\text{Ro}}_0 = 10^9, z_0/\ell_{\text{max}} = 10^{-4}$  compared to  $\widetilde{\text{Ro}}_0 = 10^9, z_0/\ell_{\text{max}} = 10^{-6}$ ).

We have added the following about the fixed stability: Note that a fixed atmospheric stability refers to a set value of a stability parameter, as opposed to a calculated stability condition that one could obtain from a transient ABL solution, which is typically the case for an ABL model including a potential temperature equation, as shown by Sogachev et al. (2012).

12. P12,19: "as an inflow model"

Adopted.

13. P17,14: "The results are stored as a function"

Adopted.

**Reviewer 2 (Fabien Margairaz)**

In this work, the authors present a new 1D model for ABL without wind veer (to be used as inflow condition for RANS). The proposed model demonstrates to satisfy Rossby and Reynolds numbers through a few 'numerical proofs'. Finally, the authors demonstrate the use of their model in a series of 1D RANS simulations aimed at wind turbine modeling. The authors also show how the model parameters can be adapted to reproduce the desired profiles.

My main comment is: I would recommend adding extra explanations at the beginning of section 2, to clarify for the reader under which atmospheric stability the model is applicable and how atmospheric stability is introduced (especially given that not potential temperature equation are considered). Otherwise, section 2 read as if the model is only valid under neutral stability and the reader might be confused to see that in section 7 stable condition is considered.

Overall, the paper is well written and only minor revisions are needed. I recommend this manuscript for publication in WES.

**Major comments:**

1. P3L30 - stability is never mentioned before, make this paragraph confusing for the reader. It your model only valid for neutral stability?

The model can be employed for both neutral and stable atmospheric stability conditions as was mentioned in the introduction and in Section 2. Reviewer 1 also pointed out that it could be clarified and we have added information in Section 2 (see answer to comment 5 of Reviewer I). 2. P9L23 - "fixed atmospheric stability" How? Maybe I am missing, I am not sure I understand the link with atmospheric stability here.

We understand the confusion, and we have therefore added the following: Note that a fixed atmospheric stability refers to a set value of a stability parameter, as opposed to a calculated stability condition that one could obtain from a transient ABL solution, which is typically the case for an ABL model including a potential temperature equation, as shown by Sogachev et al. (2012).

**Minor comments:**

1. P1L2-4 - "We propose a pressure-driven ... effects of wind veer and ABL depth." Please reformulate this sentence. Maybe expand on 'for isolating the effect of wind veer and ABL depth', as it does not illustrate the goal well.

We have reformulated: for isolating the effects of wind veer and ABL depth to to separately demonstrate the impact of wind veer and ABL depth.

2. P1L12 - Please add some references to LES and RANS.

We have added a review article about LES ABL modeling from Stoll et al. (2020) and re-referred to Apsley and Castro (1997); Blackadar (1962); van der Laan et al. (2020b) for RANS.

3. P1L12 - "Lower fidelity models"? never mentioned again, no example or reference.

We have removed this for clarity. We meant to refer to micro-scale models that represent a subset of the RANS equations, for example a linearized RANS flow model.

4. P1L17-19 - Please add some references for: "In addition, RANS can simulate ... and wake superposition."

We have added references, see answer to Reviewer I.

5. P1L22 - Please add some references for RANS model with some added or removed "components of ABL physics".

We have added a clarification, see answer to Reviewer I.

6. P2L23 - I would not use the word 'obviously' here.

Adopted, see answer to Reviewer I.

7. P2L25 - Please correct: "In the this article,"

**Adopted.**

8. P2L31 - ASL not defined (P2L4)

We have now defined ALS as Atmospheric Surface Layer.

9. P4L26 - remove 'here', as it makes this more readable

Adopted.

10. P5L1 - comma misplaced.

**Adopted.**

11. P5Eq.7 - missing punctuation at the end of the equation.

**Adopted.**

12. P5L15 - "...  $\hat{S} = -(S - G)$ . Thus, we ..."

**Adopted.**

13. P6L15 - after Eq. (12), use either "with ... as ..., ... as ..." OR "where ... is ..., ... is ..."

**Adopted.**

14. P7L13 - 'textbook' - maybe consider referencing Eq. (5).

**Adopted.**

15. P8Fig2 - the overlap of the symbols makes the figure hard to read. In addition, some of the quantity plotted are not explicitly presented (I?). I suggest adding some information in the caption of the figure.

We have added the definition of the turbulence intensity in the text, at the end of Section 5 and we have reduced the number of symbols in Figs. 2 and 3, for clarity.

16. P8L3 - "Here, we ..."

**Adopted.**

17. P9L9 - this does not read well, what 'it' refer to? Please rewrite.

We have replaced *it* by *The Reynolds number* and started a new sentence.

18. P9L12 - "... as long as ..."

**Adopted.**

19. P9L22 - "... as an inflow model for ..."

**Adopted.**

20. P11Fig4 (and Fig5) - the choice of colors for is unfortunate for the no veer ones, especially the light green, which is hard too read both on screen and printed. I suggest changing.

We like to use green and blue colors to represent neutral and stable conditions, as used in previous work. We do agree that the yellow-green color is indeed hard to see so we have changed this to a more clear light green color.

**Own** improvements**

- 1. Figure 1 and Eq. (12): We have replaced  $\xi$  with  $\eta$  for the Ellison solution since it has a different function argument compared to the Ekman solution, which we forgot to specify in the submitted version.
- 2. We have revised Fig. 1 to also include results of the ageostrophic wind speed and results for two different  $f_{pg}$  values. In addition, note that the Ekman solution in Fig. 1 from the submitted article was made using  $f_{pg} = |f_c|$  instead of  $f_{pg} = |f_c|/2$  that was reported in the caption of the submitted version.

We have added the following text about the revised version of Fig. 1:

The analytical solutions corresponding to a constant and a linear  $\nu_T$  are depicted in Fig. 1, where results of the wind speed, S, and the ageostrophic wind speed,  $\hat{S}$  are plotted. One solution with wind veer and two solutions without wind veer are shown using  $f_{pg} = |f_c|$  and  $f_{pg} = |f_c|/2$ . We have chosen to plot the results against  $z\sqrt{|f_c|/\nu_T}$  and  $2\sqrt{z|f_c|/(\kappa u_{*0})}$  to depict the differences between different values of  $f_{pg}$ . The latter would not be clear if the results are plotted against  $\xi$ , since the results for different  $f_{pg}$  would collapse for the Ekman solutions without wind veer. When  $f_{pg} = |f_c|/2$ , the ageostrophic wind speed for both Ekman solutions (with and without wind veer) are the exactly the same (Fig. 1b):  $\hat{S} = -G \exp(-\xi)$ , while the wind speed compares better if  $f_{pg} = |f_c|$  (Fig. 1a). This result may seem counter intuitive, but it simply follows from the analytic solution of Eq. (12). A similar conclusion can be made for the linear  $\nu_T$  solutions with and without wind veer (Fig. 1c and Fig. 1d), although  $\hat{S}$  is not exactly the same when  $f_{pg} = |f_c|/2$ .

**References**

- D. D. Apsley and I. P. Castro. A limited-length-scale k-ε model for the neutral and stably-stratified atmospheric boundary layer. *Boundary-Layer Meteorology*, 83:75–98, 1997.
- A. K. Blackadar. The vertical distribution of wind and turbulent exchange in a neutral atmosphere. Journal of Geophysical Research, 67(8):3095–3102, 1962.
- James Bleeg, Mark Purcell, Renzo Ruisi, and Elizabeth Traiger. Wind Farm Blockage and the Consequences of Neglecting Its Impact on Energy Production. *ENERGIES*, 11(6), 2018. ISSN 1996-1073. doi: {10.3390/en11061609}.
- V. W. Ekman. On the influence of the earth's rotation on ocean-currents. Arkiv Mat. Astron. Fysik, 2(11), 1905.
- T. H. Ellison. Atmospheric Turbulence in Surveys of mechanics. Cambridge University Press, Cambridge, U. K., 1956.
- A. Sogachev, M. Kelly, and M. Y. Leclerc. Consistent two-equation closure modelling for atmospheric research: Buoyancy and vegetation implementations. *Boundary-Layer Meteorology*, 145:307–327, 2012.
- N. N. Sørensen. General purpose flow solver applied to flow over hills. PhD thesis, Risø National Laboratory, Roskilde, Denmark, 1994.
- R. Stoll, J. A. Gibbs, S. T. Salesky, W. Anderson, and M. Calaf. Large-eddy simulation of the atmospheric boundary layer. *Boundary-Layer Meteorology*, 177(2):541–581, 2020. doi: {10.1007/s10546-020-00556-3}.
- M. P. van der Laan, N. N. Sørensen, P.-E. Réthoré, J. Mann, M. C. Kelly, N. Troldborg, K. S. Hansen, and J. P. Murcia. The k-ε-fp model applied to wind farms. *Wind Energy*, 18(12):2065–2084, December 2015. doi: 10.1002/we.1804.
- M. P. van der Laan, S. J. Andersen, M. Kelly, and M. C. Baungaard. Fluid scaling laws of idealized wind farm simulations. *Journal of Physics: Conference Series*, 1618:062018, sep 2020a. doi: 10.1088/1742-6596/ 1618/6/062018.

Figure 1: Analytic solutions with and without wind veer. (a, b) Constant  $\nu_T$ , (c, d) Linear  $\nu_T$  and  $\text{Ro}_0 = 10^5$ .

M. P. van der Laan, M. Kelly, R. Floors, and A. Peña. Rossby number similarity of an atmospheric rans model using limited-length-scale turbulence closures extended to unstable stratification. Wind Energy Science, 5(1):355–374, 2020b. doi: 10.5194/wes-5-355-2020.